# A Novel Module Promotes Horizontal Gene Transfer in *Azorhizobium caulinodans* ORS571

**DOI:** 10.3390/genes13101895

**Published:** 2022-10-19

**Authors:** Mingxu Li, Qianqian Chen, Chuanhui Wu, Yiyang Li, Sanle Wang, Xuelian Chen, Bowen Qiu, Yuxin Li, Dongmei Mao, Hong Lin, Daogeng Yu, Yajun Cao, Zhi Huang, Chunhong Cui, Zengtao Zhong

**Affiliations:** 1Department of Microbiology, College of Life Sciences, Nanjing Agricultural University, Nanjing 210095, China; 2Animal, Plant and Food Inspection Center, Nanjing Customs, No. 39, Chuangzhi Road, Nanjing 210019, China; 3Tropical Crops Genetic Resources Institute, Chinese Academy of Tropical Agricultural Science, Danzhou 571737, China; 4College of Resources and Environmental Sciences, Nanjing Agricultural University, Nanjing 210095, China

**Keywords:** horizontal gene transfer, integrative conjugative element, *Azorhizobium caulinodans*, regulation, minimal ICE*^Ac^*

## Abstract

*A**zorhizobium caulinodans* ORS571 contains an 87.6 kb integrative and conjugative element (ICE*^Ac^*) that conjugatively transfers symbiosis genes to other rhizobia. Many hypothetical redundant gene fragments (rgfs) are abundant in ICE*^Ac^*, but their potential function in horizontal gene transfer (HGT) is unknown. Molecular biological methods were employed to delete hypothetical *rgfs*, expecting to acquire a minimal ICE*^Ac^* and consider non-functional *rgfs* as editable regions for inserting genes related to new symbiotic functions. We determined the significance of *rgf4* in HGT and identified the physiological function of genes designated *rihF1a* (AZC_3879), *rihF1b* (AZC_RS26200), and *rihR* (AZC_3881). In-frame deletion and complementation assays revealed that *rihF1a* and *rihF1b* work as a unit (*rihF1*) that positively affects HGT frequency. The EMSA assay and *lacZ*-based reporter system showed that the XRE-family protein RihR is not a regulator of *rihF1* but promotes the expression of the integrase (*intC*) that has been reported to be upregulated by the LysR-family protein, AhaR, through sensing host’s flavonoid. Overall, a conservative module containing *rihF1* and *rihR* was characterized, eliminating the size of ICE*^Ac^* by 18.5%. We propose the feasibility of constructing a minimal ICE*^Ac^* element to facilitate the exchange of new genetic components essential for symbiosis or other metabolic functions between soil bacteria.

## 1. Introduction

Horizontal gene transfer (HGT) is a universal phenomenon that involves the transfer of extra genetic material from one cell to another through various mechanisms. Conjugation, transformation, and transduction are the main prevalent HGT methods [1,2,3], and other mechanisms are constantly being discovered, such as the membrane vesicle (MV) model [4,5]. HGT events exist extensively in prokaryotes and eukaryotes [6,7,8,9], thereby making numerous contributions to the ecosystem, especially adaptive evolution [10,11,12].

Rhizobia belong to various genera of α- and β-proteobacteria that interact with their legume hosts to facilitate nutrient (nitrogen) demanded by plants, and in return, plants provide energy, ultimately resulting in the formation of symbiotic organs named nodules [13,14,15]. The symbiosis genes in rhizobia that are associated with the process of nodule formation and nitrogen-fixing are generally found in extrachromosomal replicons or integrative and conjugative elements (ICEs) [16,17,18,19], which are spread mainly through conjugation, converting soil bacteria into symbionts and expanding the host range of other rhizobia [20,21,22,23]. The HGT of the symbiosis module is a significant player in bacterial evolution, biodiverse ecosystem network, material cycle, and sustainable production, promoting the diversity of rhizobia [24,25,26,27].

As gram-negative bacteria, the α-proteobacteria *Azorhizobium*
*caulinodans* have a dual capacity to fix nitrogen both under free-living conditions and in symbiosis with the legume plant, *Sesbania rostrata*, forming both root and stem nodules [28]. Our previous work revealed that *A. caulinodans* ORS571 has an 87.6-kb integrative and conjugative element (ICE*^Ac^*) containing nodulation genes, which can excise and form a circular DNA, then conjugatively transfer into various rhizobia species, allowing them to form nodules on *S. rostrata*. It interred that ICE*^Ac^* may work as a synthetic biological element carrying special functional genes and endowing recipient cells’ new physiological phenotypes. The model strain *Mesorrhizobium loti* R7A contains an enormous symbiosis island (502-kb), limiting its modification and transformation to other rhizobia or soil bacteria [29]. The process of receiving and maintaining HGT elements may increase the genetic load cost in recipient cells [30,31]. Moreover, bacteria have evolved some tactics to prevent exogenous DNA fragment into the cells, such as TraS or CRISPR/Cas9 system [32]. A study indicated that only 63 kb (58 genes) on the 1.35 Mb plasmid pSymA of *Sinorhizobium fredii* NGR234 is required for symbiosis [33], and complementary 7 nodulation genes are sufficient for nodule formation after eliminating pSymA of NGR234 [34]. In contrast, ICE*^Ac^* carries fewer native genes and provides a cleaner pedestal for gene insertion and modulation. The non-functional genes named redundant gene fragments (*rgfs*) are abundant in the ICE*^Ac^*. They may work as potential modification sites to shorten the length of ICE*^Ac^*, in order to increase the packing ratio. The smaller ICE*^Ac^* can be regarded as a superior genetic tool, which may maximize the packing capacity and carry more properties into recipient cells. Additionally, several genes in ICE*^Ac^* are required for ICE*^Ac^* transfer, such as *ahaR*, *intC*, *traB*, and *traG* [35]. Modeling these genes may increase the transfer frequency of ICE*^Ac^* or artificially control ICE*^Ac^* transfer on special location. For instance, molecular biological methods were developed to modify and assemble specific genes such as *mocB* and *nifA* in *A. caulinodans*, and engineer a plant-controlled nitrogen-fixing bacterium only when in contact with *RhiP* barley roots [36]. A xenobiotic response element (XRE) family gene containing a DUF433 domain and a set of uncharacterized genes were identified as key HGT genes in *Rhizobium etli* CFN42 and *Rhizobium leguminosarum* strain VF39SM by deleting hypothetical genes [37,38,39]. The molecular biological methods offered the possibility of deleting potential *rgfs* to explore and modify crucial genes related to ICE*^Ac^* HGT [30,31,32,33,34,40,41] in order to maintain a minimal ICE*^Ac^* genetic tool that can efficiently pack large DNA fragments and transfer interesting functional genes into recipient cells under desired conditions.

In this study, we knocked out 4 hypothetical *rgfs* from ICE*^Ac^* individually or together to shorten the length of ICE*^Ac^*. We discovered 3 non-functional *rgfs* that can serve as editable regions fitting modifications. Interestingly, we found that *rgf4* made an indispensable contribution to ICE*^Ac^* transfer. Our preliminary analysis of genes in *rgf4* revealed 3 genes that were directly correlated with the HGT process. Moreover, the mutual regulatory relationships have been elucidated and refined the regulation pathway of *intC*. In brief, we constructed a shorten transfer subset, which is conducive to the success of HGT and the exploration of crucial HGT genes and explicated a set of potential modeling proteins to promote HGT frequency. Our work will highlight synthetic remodeling thought to increase packing capacity and transferring frequency of ICE*^Ac^*, carrying more interesting gene clusters into soil bacteria.

## 2. Materials and Methods

### 2.1. Bacterial Strains Growth Conditions

Bacterial strains and plasmids and their associated characteristics are listed in Appendix A. *Azorhizobium*
*caulinodans* ORS571 and its derivative strains were grown in TY medium at 28 °C [42]. *Escherichia coli* strains were routinely grown overnight in Luria-Bertani (LB) medium at 37 °C. Antibiotics were added to the medium as needed, with the final concentration as follows: ampicillin (Amp, 100 µg·mL^−1^), gentamicin (Gm, 20 µg·mL^−1^), kanamycin (Km, 100 µg·mL^−1^), tetracycline (Tet, 10 µg·mL^−1^), chloramphenicol (Chl, 30 µg·mL^−1^), streptomycin (Str, 20 µg·mL^−1^), and spectinomycin (Spe, 100 µg·mL^−1^). The bacteria were cultured for 72 h and optical density (OD_600_) measurements were taken at different times to compare the growth capability of the strains investigated. Meanwhile, 100 µL of bacterial suspension solution were extracted, gradient diluted, and placed on TY plates at 28 °C for 3 days to count visible colonies (colony-forming units, CFU). In addition, β-d-1-thiogalacto-pyranoside (IPTG) with a final concentration of 100 µg·mL^−1^ was added into the medium to induce gene expression when the strains containing recombinant plasmid pSRKGm.

### 2.2. Bioinformatics Analyses

Analysis was carried out of how relevant genes are organized in various genomes using an online website (http://www.microbesonline.org/, accessed on 1 July 2022). The online website (https://www.ncbi.nlm.nih.gov/genome/, accessed on 1 August 2022) provided the genomic information listed below: *Azorhizobium*
*caulinodans* ORS571 (NC_009937.1), *M*. *japonicum* R7A (CP033366.1/CP051772.1), *Mesorhizobium* sp. BNC1 (NC_008254.1), *Bradyrhizobium* sp. BTAi1 (NC_009485), *Bradyrhizobium japonicum* USDA110 (NC_004463), *Rhizobium* sp. CIAT894 (CP020947.1), *Paracoccus denitrificans* PD1222 (NC_008687), *Nitrobacter winogradskyi* Nb-255 (NC_007406) and *Xanthobacter autotrophicus* Py2 (GCA_000017645.1).

The protein sequences of *rgf4* were subjected to a multiple sequence alignment tool (https://www.ebi.ac.uk/Tools/msa/clustalo/, accessed on 1 August 2022) and protein homologues were identified by BLASTP analysis (https://blast.ncbi.nlm.nih.gov/Blast.cgi, accessed on 1 August 2022). Protein sequences were aligned and visualized in Geneious v. 2020.0 (Biomatters, Auckland, New Zealand) using the Clustal Omega algorithm.

Genes promoters were predicted by an online website (http://www.softberry.com/, accessed on 1 July 2022). We used the MEME Suite [43] to predict potential RihR binding motifs in the promoter regions of genes adjacent to *rihR*. The Tomtom program was employed to find similar motifs in published libraries.

### 2.3. Molecular Biology Techniques

Appendix A shows primers used to amplify corresponding DNA fragments. Appendix A describes the construction of recombinant plasmids and the process of homologous recombination. Homologous fragments and Km fragment were cloned into a suicide vector pEX18Gm containing a *sacB* selectable marker [44,45], and the recombination plasmid was introduced into *E. coli* SM10 λpir as donor strain [46,47]. The plasmid was transferred from the donor into ORS571 by biparental conjugation. Donor and recipient cells were mixed in a 1:1 ratio and placed on a filter membrane, which was placed on TY solid medium to cultivate for 12 h at 28 °C. The cells mixture was evenly distributed on TY agar containing Km/Gm to screen conjugation colonies, which was the first homologous recombination. Subsequently, the colonies were streaked on TY plates to generate double-crossover events. TY-sucrose (10% sucrose) plates were used to exclude strains that stayed in the first homologous recombination state. The candidate strains were selected on TY supplemented with Km or Km/Gm at 28 °C for three days. The positive colonies that only grew on TY plates containing Km were verified by sequencing. In the same way, other in-frame deletion strains (Appendix A) were constructed on the background of Azc1 [35] and Azc2, as follows: Δ*rgf1* (1.9 kb was deleted in Azc1), Δ*rgf**2* (4.5 kb was deleted in Azc1), Δ*rgf3* (9.2 kb was deleted in Azc1), Δ*rgf4* (2.3 kb was deleted in Azc1), Δ*rgf12* (6.4 kb was deleted in Azc1), Δ*rgf123* (15.6 kb was deleted in Azc1), Δ*rihR*, Δ*rihF1*, Δ*rihF2* and Δ*rihR**. Because pVIK112 had a Km-resistant gene, the same as Azc1, we reconstructed a new tetracycline-resistant gene insertion strain, Azc2. In addition, we confirmed the deletion strains by amplification of the coding sequence of the genes (Appendix A). For complementation, the complemented genes were amplified by PCR. These fragments were inserted into the downstream of the pSRKGm promoter using the digestion sites of restriction enzymes. The recombinant plasmids were sequenced and transferred into corresponding mutants by electroporation. When β-d-1-thiogalacto-pyranoside (IPTG) was added, the complemented genes were induced through the activation of the pSRKGm promoter.

### 2.4. Measure HGT Frequency of ICE^Ac^

Donor strains (Wild-type ORS571 [48] and its derivatives) and recipient strains (*M. huakuii* 93 [35]) were employed to determine the HGT frequency of ICE*^Ac^* as previously reported [35]. Strains were grown overnight in a liquid medium, and 2 mL bacterial suspension were mixed in a 1:1 ratio, with starting optical density at 600 nm (OD_600_ = 1.0). Before mixing with the donor, recipient bacteria were graded and diluted on TY plates. The mixture was placed on a TY solid medium to cultivate for 12 h at 28 °C and then overlaid on TY plate containing Spe/Km or Spe/Tet. The colony-forming units (CFU) of transconjugants and recipient strains were counted to calculate HGT frequency. The experiment was repeated at least three times.

### 2.5. Measuring Transcriptional and Translational Expression of Genes

The construction of transcriptional fusion reporters and β-galactosidase activity assay followed the protocol as described previously [49,50]. The transcriptional regions and promoters of target genes were respectively cloned into expression plasmid pVIK112 and pRA302 [51,52], and the plasmids were integrated into the target genes locus by homologous recombination to indicate the intensity of gene expression in the chromosome. The 200 µL cultures were extracted to detect enzyme activity. The reaction time (Δt) was recorded, and the reading of OD_420_ was measured using ο-Nitrophenyl-β-d-galactopyranoside (ONPG) as the reaction substrate. The activity was normalized to the OD_600_ of the cultures and expressed in Miller units, $\mathrm{Miller}\;\mathrm{units}\;=1000\times {\mathrm{OD}}_{420}/\left({\mathrm{OD}}_{600}\times ∆t\times 0.2\right).$ One Miller unit of enzyme activity is defined as the amount of enzyme required to generate 1 mmol ο-Nitrophenyl (ONP) per minute. Three biological replicates were performed, and the experiment was repeated at least three times.

### 2.6. Bacterial One-Hybrid Assay

The bacterial one-hybrid (B1H) assay was an efficient method to detect the specific interaction between transcriptional regulators and target genes [53,54]. Candidate gene promoters were amplified and cloned into pBT-derived reporter vector pBXcmT. The coding region of *rihR* was cloned into pTRG to acquire recombination plasmid pTRG-*rihR*. The host strain for propagation was *E. coli* XL1-Blue MRF’ Kan [55], and *rihR* expressed in pTRG interacts with bait DNA in pBXcmT. Positive cotransformants grow well on selective medium containing 10 mM 3-amino-1,2,4-triazole (3-AT), 10 µg·mL^−1^ streptomycin (Str), 10 µg·mL^−1^ tetracycline, 30 µg·mL^−1^ chloramphenicol, and 50 µg·mL^−1^ kanamycin. All medium plates were incubated at 28 °C for 3 days. Cotransformants containing vectors pTRG and pBXcmT served as negative controls, and the strain cotransformed with pTRG-R3133 and pBXcmT-R2031 was employed as a positive control [53]. The assay was performed three times, and three replicates of different strains were examined in each time.

### 2.7. Expression and Purification of His-Tagged RihR

The reading region of *rihR* was cloned into pET28a (Novagen, Madison, WI, USA) to express His-tagged RihR in *E. coli* BL21 (DE3) [46] following standard procedures [42,56,57]. *E. coli* BL21 carrying pET28a-*rihR* was grown in 500 mL LB medium with the appropriate antibiotic and then induced with the addition of 0.5 mM β-d-1-thiogalacto-pyranoside (IPTG) at OD_600_ ≈ 0.6. Cells were incubated at 16 °C for 12 h and harvested by centrifugation at 4 °C. The cells were suspended in a pH 8.0 lysis buffer (50 mM NaH_2_PO_4_ and 300 mM NaCl) and sonicated to obtain cell lysate, which was centrifuged at 4 °C for 30 min to exclude insoluble sedimentation. The soluble proteins were loaded onto the Ni-NTA column (GE Healthcare, Chicago, IL, USA). Firstly, the column was washed with several column volumes of purification buffer (lysis buffer plus 20 mM imidazole). Subsequently, N-terminally His-tagged RihR was eluted with elution buffer (lysis buffer plus 250 mM imidazole) and assessed by SDS-PAGE. The purification was desalted by HiTrap Desalting Column (GE Healthcare, Chicago, IL, USA), and protein concentration was determined by the NanoDrop 2000c (Thermo Fisher Scientific, Waltham, MA, USA).

### 2.8. Electrophoretic Mobility Gel Shift Assay (EMSA)

The detailed protocol of EMSA has been described in previous research [58,59]. Briefly, approximately 300 bp promoter fragment of the candidate gene was amplified from *A. caulinodans* ORS571 genome and purified as a DNA probe using the PCR purification kit (Sangon Biotech, Shanghai, China). The DNA probes (50 ng) were mixed in a binding buffer containing different concentrations of *RihR* protein in a final volume of 20 µL. The binding buffer contained 50 mM Tris-HCl (pH 8.3), 0.25 M KCl, 2.5 mM dithiothreitol (DTT), 5 mM MgCl_2_, 0.25 mg·mL^−1^ bovine serum albumin, 0.05 mg·mL^−1^ poly(dI-dC), 2.5 mM EDTA, 1% glycerol. The reaction mixtures were incubated for 20 min at room temperature and then electrophoresed on a 6% nondenaturing polyacrylamide gel in 0.5× Tris-borate-EDTA buffer at 150 V for 70 min. Gels were stained and photographed using GelRed (Sangon Biotech, Shanghai, China) and the molecular imager Gel Doc XR system (Bio Rad, Hercules, CA, USA).

### 2.9. Quantitative Real-Time PCR Analysis

qRT-PCR was used to measure the transcriptional level of genes *rihF1*, *rihR*, and *ahaR*. Total RNA was extracted using TRIzol method [60]. A cDNA Synthesis Kit (Vazyme Biotech, Nanjing, China) was used to convert RNA into cDNA using specific amplified primers (Appendix A), following the manufacturer’s instructions. The qPCR experimental program was described below: 30 s at 95 °C, followed by 40 cycles of 10 s at 95 °C and 30 s at 60 °C, and finally, 72 °C (30 s), according to the manufacturer’s protocol for the SYBR green detection Kit (Vazyme Biotech, Nanjing, China). Three independent experiments were performed (each in triplicate), the transcript levels were normalized by endogenous control 16S rRNA [61], and the relative expression levels of target genes were calculated by the Comparative CT method [62].

### 2.10. Statistical Analysis

GraphPad Prism 8.0 software (GraphPad, La Jolla, CA, USA) was applied to analyze and draw the experimental data, which at least referred to three biological samples and three independent experiments. All data were presented as mean values with standard deviations (±SD) indicated by error bars. Student’s *t*-test was used to compare two groups of data and calculate the *p* values, and *p* values < 0.05 were considered significant. One-way ANOVA with a post hoc Tukey-Kramer test of honestly significant difference was also applied in data analysis.

## 3. Results

### 3.1. Identification of rgfs Involved in ICE^Ac^ Transfer

Following our previous data and bioinformatics analysis [35], 4 hypothetical *rgf* regions were marked on the *A. caulinodans* symbiosis island (Figure 1A). In order to confirm whether these *rgfs* were involved in ICE*^Ac^* transfer, we performed single or multi *rgfs* knockout. We found Δ*rgf1*, Δ*rgf2*, Δ*rgf3*, Δ*rgf12*, and Δ*rgf123* were consistent with that of WT strain on HGT frequency (Figure 1B). Surprisingly, a 100-fold decrease in the HGT frequency of Δ*rgf4* was observed. Additionally, the growth capacities of all the mutants were comparable to that of the WT strain (Appendix A). Collectively, these data suggest that genes within or adjacent to *rgf4* were vital for ICE*^Ac^* transfer. In conclusion, we successfully shortened the ICE*^Ac^* length by approximately 18.5% without affecting the transfer frequency by deleting *rgf123*, and these regions can be used as sites to insert editable interesting genes to remodel ICE*^Ac^*. 

### 3.2. Genomic Context of rgf4 Genes in Different Strains

In the *rgf4* fragment (Figure 2A), a transposase gene, AZC_3878, was not associated with ICE*^Ac^* transfer [35]. We hypothesized that other genes in *rgf4* serve as relevant ICE*^Ac^* HGT-related genes, respectively named them *rihF1a* (AZC_3879), *rihF1b* (AZC_RS26200), and *rihF2* (AZC_3880). Bioinformatics analyses suggested that AZC_3879 and AZC_RS26200 shared the same promoter, which may be expressed as a unit named *rihF1* (AZC_3879 and AZC_RS26200). Interestingly, the homologues of *rihF1a*, *rihF1b*, *rihF2*, and AZC_3881 are prevalent in different strains (Figure 2B,C), implying that these genes may serve as a conserved module in controlling the frequency of HGT. Moreover, XRE family proteins usually regulate their downstream genes to control various metabolic pathways [63,64,65]. Thus, we assumed that AZC_3881 might encode a relevant ICE*^Ac^* HGT regulation protein named *rihR*. The RihF1a, RihF1b, RihF2, and RihR, respectively, share 35.35%, 33.80%, 46.05%, and 38.24% sequence homologies with the genes are required for ICEMlSym^R7A^ transfer [66,67], which suggested the possibility that the module has significance for HGT. Furthermore, we statistically found that the module is distributed diversely in different ICEs, such as the symbiosis island ICEMlSym^R7A^ (inside ICE) and the *B*. *japonicum* USDA110 symbiosis island (outside ICE) [68]. The uncertain distribution may be the result of gene rearrangement. On the other hand, HGT-related genes (marked with yellow) are conservative located in the upstream or downstream of the module and belong to different strains (Figure 2B), such as *traG* and *traF*. In previous work, we proved that AZC_3882 (*intC*) is critical for ICE*^Ac^* HGT frequency [35]. We, therefore, hypothesized that the module could interact with adjacent *intC* to maintain ICE*^Ac^* HGT.

### 3.3. rihF1 and rihR Contribute to the Conjugation Frequency of ICE^Ac^

We individually deleted genes in the module in order to determine which gene contributes to ICE*^Ac^* transfer. The growth curve experiment showed that mutant strains had no difference with WT stain Azc1 (Appendix A). Δ*rihF1* and Δ*rihR* stains showed approximately 1000- and 100-fold decreases in the HGT frequency, respectively (Figure 3A). In addition, these decreases were restored to WT levels following complementation with *rihF1* and *rihR* (Figure 3B,C). These data supported that *rihF1* and *rihR* are crucial genes relevant to ICE*^Ac^* HGT frequency. However, the identical phenotype was not observed in Δ*rihF2* (Figure 3A), indicating that *rihF2* is a non-functional gene in the process of ICE*^Ac^* HGT. We further investigated the individual contribution of *rihF1a* and *rihF1b* to ICE*^Ac^* HGT. We introduced *rihF1a* and *rihF1b* into Δ*rihF1* separately, and the results showed that the HGT frequency of each complementary stain remained the same ratio as that of Δ*rihF1* strain, lower than that of the Azc1^Vec^ strain (Figure 3D), verifying the hypothesis that *rihF1a* and *rihF1b* work as a unit (*rihF1*) to maintain ICE*^Ac^* transfer. These results suggested that *rihF1* and *rihR* are essentially required for ICE*^Ac^* HGT.

### 3.4. ICE^Ac^ HGT Is Influenced by rihF1 and rihR as Two Indipendent Pathways

To further investigate how this module works, we first studied whether RihR regulates *rihF1* or *intC* to affect conjugation frequency. We used the MEME motif discovery platform to identify shared sequence motifs in the promoter regions of *rihF1a*, *rihF1b*, *rihF2*, *rihR*, and *intC*, expecting to obtain a possible target sequence for RihR. The promoter sequences were submitted to MEME (Figure 4A) and showed a motif with high similarity to the HTH-type regulator RutR [69,70]. Moreover, the predicted motif was present in the *intC* and *rihF1b* promoter sequences (Figure 4B), suggesting that these two genes may be regulated by RihR. To determine the regulation relationship, the recombinant plasmids pRA302-*intC::lacZ* and pVIK112-*rihF1::lacZ* were introduced into Δ*rihR** and Azc2 strains separately (Appendix A). Subsequently, the expression of *rihF1* and *intC* was measured through a *lacZ*-based reporter system, respectively (Figure 4C). As indicated, Azc2 showed significantly higher *intC* expression activity than that of the Δ*rihR** strain(~2-fold), suggesting that RihR positively regulates *intC* but has no regulation effect on *rihF1*. Consequently, *rihR* and *rihF1* maintain the conjugation frequency through different pathways, and the *rihR* performs its function by influencing *intC* expression.

### 3.5. RihR Directly Binds to the Promoter Region of IntC

The EMSA and B1H assay were used to investigate whether RihR protein directly regulates *intC*. The primers used for promoter fragment amplification are listed in Appendix A. RihR was expressed in BL21 (DE3) as a 6× His-tagged fusion protein (~14 kDa) and purified using a Ni-NTA resin column (Figure 5A). A protein-DNA complex was observed in different lanes, which generated different heights of shift with increasing concentrations of the purified RihR protein (Figure 5B). As controls, RihR did not bind to a 160 bp non-specific promoter (lanes 6 and 7). This result indicated that RihR specifically binds to the *intC* upstream sequence directly in vitro. 

We employed the B1H assay to further verify the EMSA conclusion [53]. The cotransformant strain, containing pTRG-*rihR* and pBXcmT-*intC*, could be survived on the screening medium containing 3-amino-1,2,4-triazole (3AT) and streptomycin (Figure 5C), which suggests that regulator protein, RihR, interacts with the *intC* promoter on pBXcmT and initiates the expression of reporter genes. These results further support that RihR directly binds on the *intC* promoter to induce *intC* expression, which positively affects ICE*^Ac^* transfer.

### 3.6. Regulation of rihR Is Independent of AhaR

Our previous results demonstrated that the transcription factor, AhaR, positively regulates *intC* gene expression by binding on the promoter region, and the regulation was dependent on the plant flavonoid naringenin (NAR) [35]. We then investigated whether there exists a relationship between *ahaR* and *rihR* when they regulate *intC* expression. The *rihR* expression in Azc1 and Δ*ahaR* was measured, and the result showed that there had been no difference on *rihR* expression between WT and Δ*ahaR* strain at different conditions, with or without the presence of NAR (Figure 6A). Consequently, AhaR had no regulatory impact on *rihR*. The qRT-PCR analysis also indicated that the expression of *rhiF1* was not induced by AhaR (Appendix A), revealing that *rihF1* maintained ICE*^Ac^* transfer independently of *ahaR* and *rihR*. These data showed that AhaR and *rihR* regulated *intC* separately, and AhaR did not regulate *rihF1* to influence ICE*^Ac^* HGT. In turn, we estimated whether RihR could regulate *ahaR* to influence ICE*^Ac^* transfer. The result of *ahaR*-*lacZ* activity confirmed that *ahaR* was not regulated by RihR (Figure 6B); the qRT-PCR analysis also clarified the result (Appendix A). Collectively, the aforementioned data supported the idea that *rihR* and *ahaR* respectively regulated *intC* expression to maintain and enhance ICE*^Ac^* transfer. In order to maintain ICE*^Ac^* transfer, ORS571 employs a complex regulation pathway and promotes ICE*^Ac^* transferring into soil bacteria, broadening the host range of rhizobia.

## 4. Discussion

HGT plays an important role in bacteria evolution [71], sharing beneficial gene cluster among bacteria, such as symbiosis genes [22,23]. However, the transfer of macromolecular symbiotic islands may be an additional burden for the host cell [11], which makes the deletion of *rgfs* necessary. 

In a successful instance of synthetic biology, the *nif* operons from *Klebsiella pneumoniae* are integrated into E. coli after deleting redundant genes, giving it equal nitrogen-fixing activity [72]. ICE*^Ac^* also contains a large number of redundant genes. Gene knockout offers an established approach for the study of unknown function *rgfs* [73]. We deleted the hypothetical *rgfs* and identified 3 genes located on *rgf4* required for ICE*^Ac^* transfer through the gene deleting method. Excision of the *rgf123* fragment reduced the ICE*^Ac^* size by 18.5%, discovering a 15.6-kb gene editable region (Figure 1A,B). Meanwhile, the construction of minimal ICE*^Ac^* can excavate more crucial genes relevant to HGT. In a classic case, the nitrogenase activity of *A. caulinodans* shows no ammonium repression through the modification of *nifA* gene [74]. Thus, these crucial HGT genes can be further modified to enhance the conjugation frequency of ICE*^Ac^*. The bacteria, such as *Ralstonia solanacearum*, transferred with nodulation and nitrogen-fixation genes can provide both themselves and hosts with competitive advantages [75,76,77]. These findings show that the transfer of functional genes is vital for bacteria in nature. For the subsequent study, modifiable regions in ICE*^Ac^* could be substituted by nitrogen fixation and other functional genes, conferring more advantage phenotypes to recipient cells. The regions can also be replaced by reporter genes such as green fluorescence protein, which can assist in the observation of ICE*^Ac^* transfer into diverse soil bacteria in different environments [78]. We expect to develop a remodeling ICE*^Ac^* genetic tool that can pack more functional gene clusters into soil bacteria.

The protein sequence alignment suggests that the module located on *rgf4* and *rihR* may be widespread in bacteria (Figure 2A–C), participating in HGT regulation. We have shown that *rihR* and *rihF1* in *A. caulinodans* are engaged in the HGT process (Figure 3), but the module in other strains requires further experimental investigation. Only one of the strains, *M. loti* R7A, with a homologous module, has been reported to promote ICEMlSym^R7A^ HGT [66,67]. The +1 programmed ribosomal frameshift (PRF) fuses two TraR-activated genes, *msi172* and *msi171*, producing an activator FseA to promote HGT. The HGT process is mediated by the regulation of quorum-sensing (QS) gene *traR*. Conversely, the homologous groups of *msi172* and *msi171* in *A. caulinodans*, *rihF1b*, and *rihF1a* contain individual start and end codons themselves. Thus, the PRF phenomenon and *traR* are absent in *A. caulinodans* [79,80,81]. These data indicate that *rihF1* contains a different regulation mechanism from *M. loti* strain R7A to maintain ICE*^Ac^* transfer. *rihF1* and *rihR* may work as potential remodeling sites to promote ICE*^Ac^* transfer frequency.

In this study, we improved a new regulation feature of *intC*. In brief, *intC* can be expressed at a basal level regulated by a new regulon RihR; it also is positively regulated by AhaR sensing host signal, NAR. A previous study has proven that the XRE-family genes located among or nearby HGT-related genes are required to conjugative HGT [37,38]. In *A. caulinodans*, the XRE-family RihR plays a significant role in maintaining ICE*^Ac^* transfer by promoting *intC* expression (Figure 4). The response to plant-exudation of soil bacteria is a key parameter through which bacteria control HGT in rhizosphere [82,83]. AhaR responds to the plant inducer NAR and promotes *intC* expression to enhance HGT [35], but no regulatory relationship exists between *ahaR* and *rihR* (Figure 6). Thus, we enrich our previous study results by finding that there exists a complex regulation pathway of *intC* expression. In *Xanthomonas oryzae* MAFF311018, GamR employed two independent regulators, HrpG and HrpX, to regulate *hrp* expression [84]. We found that *A. caulinodans* has developed multiple pathways to regulate ICE*^Ac^* transfer through a long period of evolution. In the absence of NAR, *rihR* and *rihF1* serve as two key components to maintain the general level of HGT. In the presence of NAR, the HGT level is enhanced by *ahaR* (Figure 7). This study offers strong evidence that *A. caulinodans* has evolved multiple strategies to maintain or enhance HGT under different environmental conditions, and the complex mechanism remains to be further investigated. Our work contributes to the preliminary elucidation of the vital HGT regulation pathways in *A. caulinodans*, which may be applied in ICE*^Ac^* remodeling. Meanwhile, our works highlight the thought that ICE*^Ac^* may work as a new bacterial remodeling element through increasing package ratio and transferring frequency by deleting *rgfs* and embellishing special genes. 

## Figures and Tables

**Figure 1 genes-13-01895-f001:**
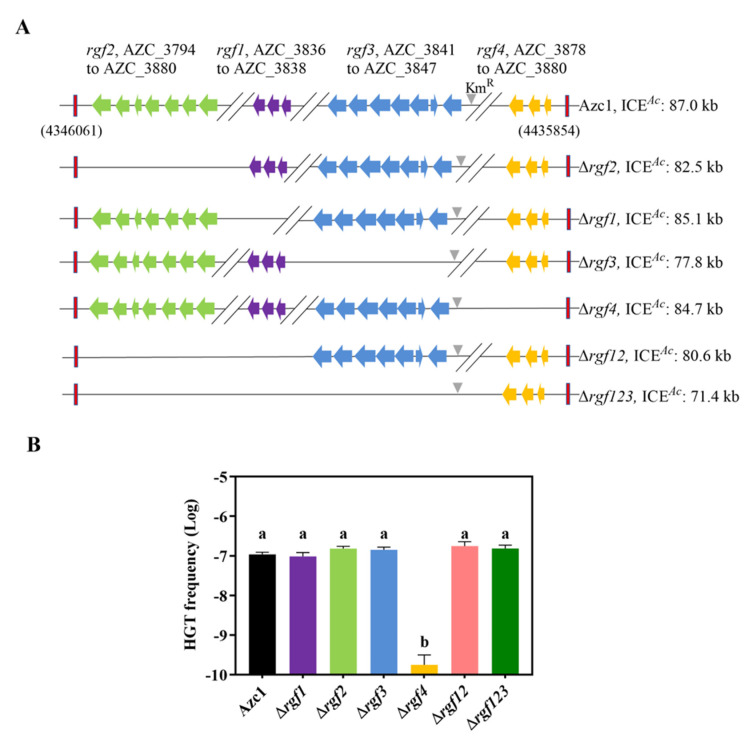
Differential consequences of *rgf* mutants on ICE*^Ac^* HGT frequency. (**A**) Distribution of 4 hypothetical *rgf* regions in ICE*^Ac^*, with a deletion strategy starting from the second row down. Red rectangles represent the entire ICE*^Ac^* region. The grey triangle represents the kanamycin-resistant gene, and other colors indicate different hypothetical *rgfs*. (**B**) Identification of *rgf4* is required for ICE*^Ac^* HGT frequency. Azc1 (wild-type) or mutants were mixed with *M. huakuii* 93 at 28 °C, respectively. The colony-forming units (CFUs) of transconjugants and the recipient were determined to calculate the HGT frequency. Data are mean ± SD of three independent experiments. Means not connected by the same letter are significantly different (*p* < 0.05).

**Figure 2 genes-13-01895-f002:**
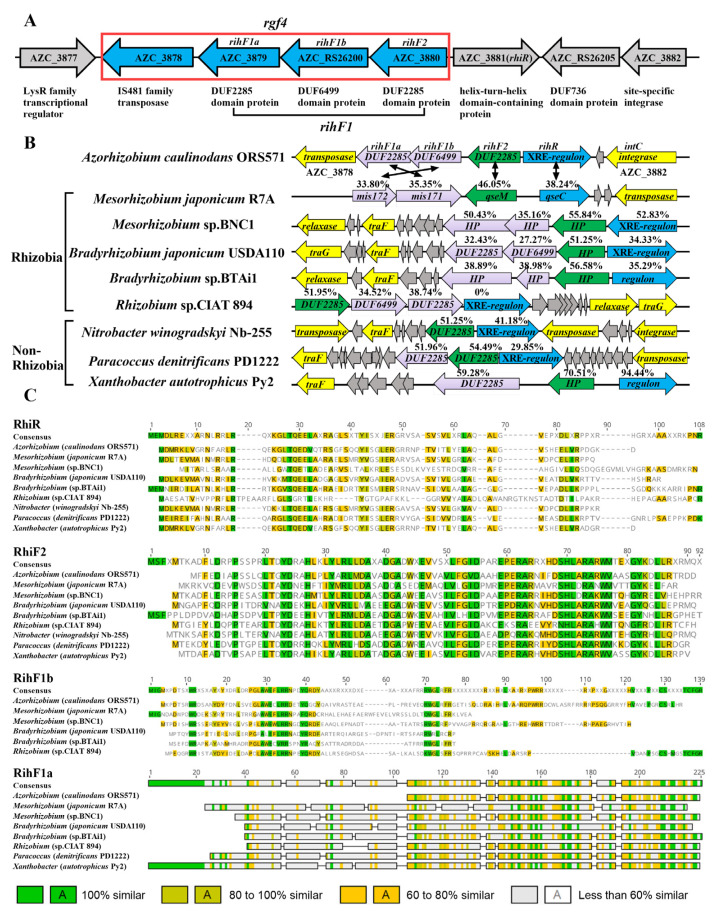
Composition and distribution of *rgf4* genes in different strains. (**A**) The genetic structure of *rgf4* in *A. caulinodans*. (**B**) The organization of *rgf4* in different strains. Inside of the arrows showed gene function. HP, hypothetical protein. Protein sequence similarity in different strains compared with *A. caulinodans* is marked above the arrows. (**C**) Multiple sequence alignments of homologous *rgf4* from various strains. Higher similarity is indicated by a darker gray backdrop for the amino acid letters.

**Figure 3 genes-13-01895-f003:**
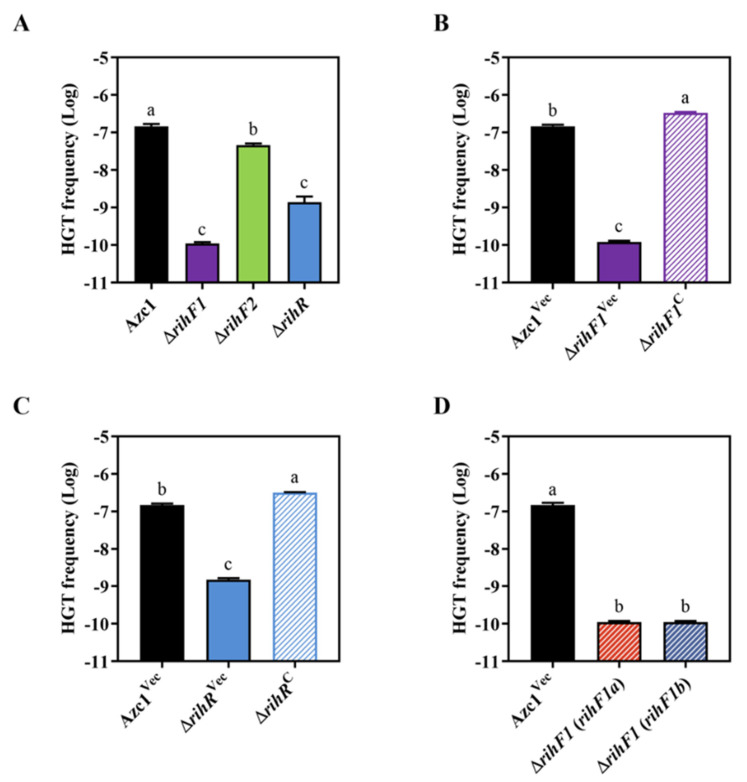
Identification transfer capacities of ICE*^Ac^* in different *rgf4 gene* deletion strains. (**A**) The transfer frequency of ICE*^Ac^* in strains Δ*rihF1*, Δ*rihF2*, and Δ*rihR*. (**B**) The transfer frequency of ICE*^Ac^* in Δ*rihF1*^Vec^ and Δ*rihF1*^C^. (**C**) The transfer frequency of ICE*^Ac^* in Δ*rihR*^Vec^ and Δ*rihR*^C^. (**D**) The transfer frequency of ICE*^Ac^* in Δ*rihF1* (*rihF1a*) and Δ*rihF1* (*rihF1b*). The strains were mixed with *M. huakuii* 93, and the CFU of the transconjugants was determined. Data are mean ± SD of three independent experiments. Means not connected by the same letter are significantly different (*p* < 0.05). The same letter indicates no significant difference (*p* > 0.05).

**Figure 4 genes-13-01895-f004:**
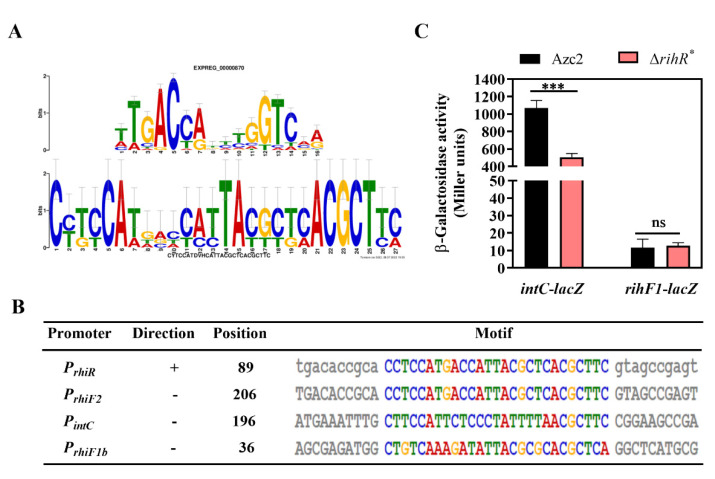
RihR positively regulates *intC* expression in vivo. (**A**) The promoter motif was predicted by MEME in *rgf4* genes and *intC*. Top: The motif results from a comparison with a known motif database by using the Tomtom program. Bottom: The motif is predicted based on the *intC*, *rihR*, *rihF2*, *rihF1a*, and *rihF1b* promoters. (**B**) The specific positions of the predicted motif in different promoters. +: The motif site was found in the sequence as it was supplied. −: The motif site is found in the reverse complement of the supplied sequence. (**C**) β-galactosidase activities of the transcriptional fusion *rihF1*-*lacZ* and the translational fusion *intC*-*lacZ* were measured in Azc2 and Δ*rihR**. Data are mean ± SD of three independent experiments. ***: Student’s *t*-test *p* < 0.001, ns: no significant difference.

**Figure 5 genes-13-01895-f005:**
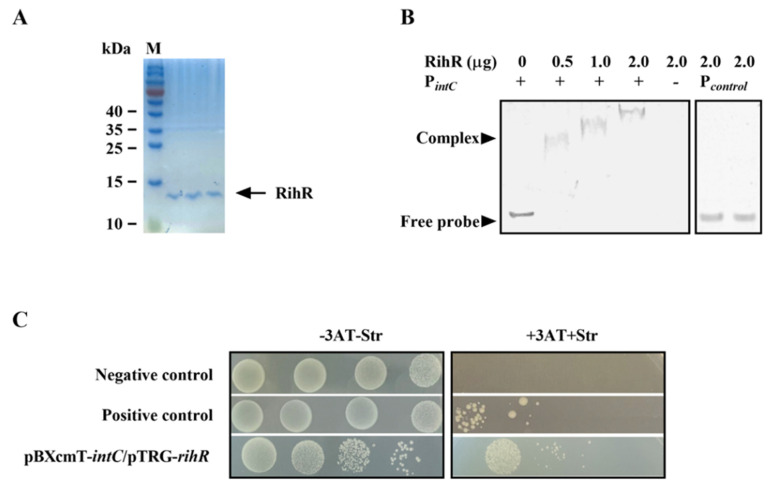
RihR binds on the promoter region of *intC* with high affinity. (**A**) The purified RihR protein was analyzed through non-reducing SDS-PAGE. The soluble proteins in cells were washed down by the elution buffer and added to the three lanes. The arrow indicates the location of RihR protein. M: marker. (**B**) EMSA assay, RihR protein binds on the *intC* promoter. The DNA fragment containing the *intC* promoter was incubated with increasing concentrations of RihR protein at room temperature. The concentration of DNA fragments in the reaction system was 50 ng. The mixture was separated on polyacrylamide non-denaturing gel to observe the DNA-protein complex. (**C**) The B1H assay of the interaction between RihR and *intC* promoter. Co-transformant strains containing the plasmids pTRG-R3133, pBXcmT-R2031 and pTRG, pBXcmT served as positive control and negative control, respectively.

**Figure 6 genes-13-01895-f006:**
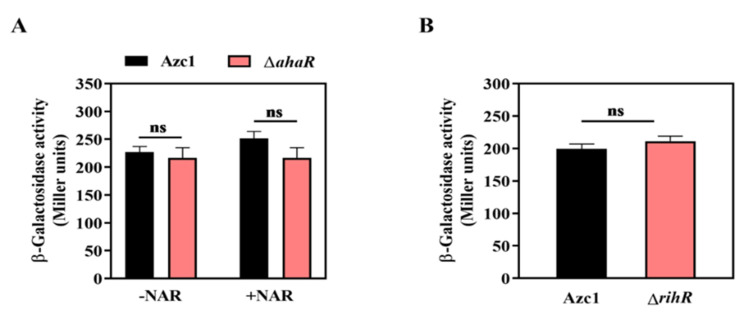
The regulatory relationship between *rihR* and *ahaR* of *A. caulinodans*. (**A**) The β-galactosidase activities of the translational fusion *rihR*-*lacZ* was determined in Azc1 and Δ*ahaR*. When indicated, 20 μM NAR was added to the medium. (**B**) The β-galactosidase activities of the translational fusion *ahaR*-*lacZ* were measured in Azc1 and Δ*rihR*. Data are mean ± SD of three independent experiments. ns: no significant difference.

**Figure 7 genes-13-01895-f007:**
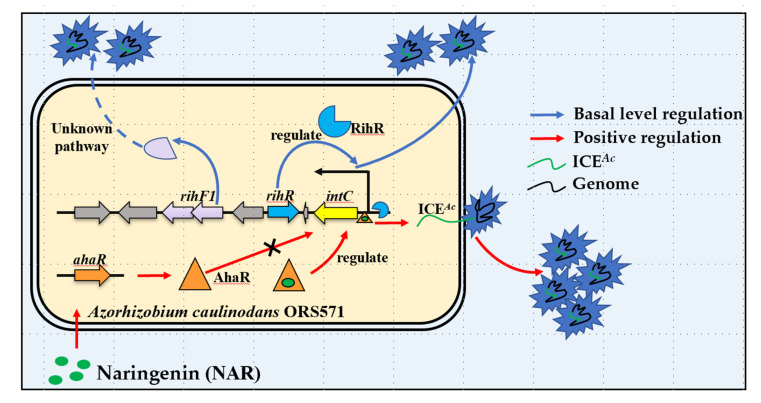
Model of ICE*^Ac^* transfer. Efficient transfer pathway: The host plant (*S. rostrata*) produces a flavonoid signal (NAR) that is recognized by AhaR. The AhaR protein is activated, then binds on the *intC* promoter to promote HGT frequency (~10^−4^). General transfer pathway: The RihR protein binds on the *intC* promoter and maintains the general level of HGT frequency (~10^−7^), and RihF1 maintains the same HGT frequency through an unknown process.

## Data Availability

Not applicable.

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
