# Peer review of "A Novel Module Promotes Horizontal Gene Transfer in Azorhizobium caulinodans ORS571"

_genes, 2022, doi:10.3390/genes13101895_

Round 1

Reviewer 1 Report

Dear Editor,

The authors describe results on “A Novel Module Promotes Horizontal Gene Transfer in Azo- 2 Rhizobium caulinodans ORS571”. Although the authors have done a good piece of work but some parts of the manuscript need minor revision again.

Comments:

        i.            In Introduction, various grammatical errors are highlighted. References are written before the start of line. These could be shifted at the end of line.

      ii.            Methodology can be shortened, if possible., In isolation of rhizobacteria and halo tolerance assay, the techniques should be supported by references

    iii.            In results, methodology lines should be avoided.

    iv.            Legends of respective figures can be revised.

Reviewer 2 Report

Comments on manuscript “A Novel Module Promotes Horizontal Gene Transfer in Azorhizobium caulinodans ORS571” by Li et al.

 In this manuscript, the authors explored the potential of 87.6 kb integrative and conjugative element in horizontal gene transfer. Authors show the importance of rgf4 genes in horizontal gene transfer and work out its partial regulation through intC. I have following specific comments on this piece of work.

1.        Introduction: Authors write “synthetic biology method ………”. Synthetic biology method is not a standard term. Authors should elaborate what was exactly done.

2.        Why do authors infer “Collectively, these data suggest that genes within or adjacent to rgf4 were vital for ICEAc transfer.” if rgf123 deletion mutant shows HGT frequency similar to WT?

3.        How HGT efficiency was calculated? How is it different from HGT frequency? If both are same, please use any one term.

4.        Please elaborate the rationale behind using the backgrounds of Ac1, Azc1 and Azc2 for the benefit of the readers. What are the genotypic differences between these strains? Why different deletion mutants were generated using different backgrounds?

5.        Fig 2C: Authors should provide a colour figure that would be visually appealing.

6.        Fig. S1B needs more details in the legend. Please mark “up” and “down” fragments in both the plasmid construct and the genome.

7.        Confirmation of deletion mutants: Deletion of appropriate genes needs to be demonstrated by showing that the gene product is absent at transcript and/ or protein level in each mutant.

8.        Composition of lysis buffer?

9.        Line 155: “the coding regions of complemented genes”. Please add specific details.

10.    Line 171: “internal regions of target genes”. Please add specific details.

11.    Please mention length of the promoter fragment used for EMSA.

12.    M. huakuii 93 should be listed in Table S1.

13.    pET28a-rihR should be listed in Table S1. Restriction sites used for all the clonings should be mentioned.

14.    Fig. 5A and B are highly adjusted for brightness and contrast making it difficult to visualize all the bands. Authors should provide better images. In Fig 5A, the target protein appears degraded as the smear spreads from 15 kDa to 10 kDa. Authors should offer some explanation in the manuscript. What is loaded in different lanes in not mentioned. EMSA image should have one more lane showing release of the probe after the treatment of DNA-protein complex with proteinase K. Give details of non-specific promoters used along with their fragment sizes and DNA sequences.

15.    What do authors mean that “we improved a new regulation feature of intC”?

16.    In Fig. 7, HGT frequency should be in positive or negative? Why ICEAC is in red circle? What do blue shapes with or without red circles indicate?

17.    Line 356 starts with [47]. Please correct.
